# Maize Cultivation in Sun Mushroom Post-Harvest Areas: Yield, Soil Chemical Properties, and Economic Viability

**DOI:** 10.3390/plants14071097

**Published:** 2025-04-02

**Authors:** Lucas da Silva Alves, Matheus Rodrigo Iossi, Cinthia Elen Cardoso Caitano, Wagner Gonçalves Vieira Júnior, Pedro Afonso Gomes Teixeira, Reges Heinrichs, Arturo Pardo-Giménez, Diego Cunha Zied

**Affiliations:** 1Graduate Program in Agricultural and Livestock Microbiology, School of Agricultural and Veterinarian Sciences, São Paulo State University (UNESP), Jaboticabal 14884-900, SP, Brazil; silva.alves@unesp.br (L.d.S.A.); matheusiossi1@gmail.com (M.R.I.); cinthia.cardoso@unesp.br (C.E.C.C.); vieira.jr@unesp.br (W.G.V.J.); afonso.gomes@unesp.br (P.A.G.T.); 2Department of Crop Production, School of Agricultural and Technological Sciences, São Paulo State University (UNESP), Dracena 17915-899, SP, Brazil; reges.heinrichs@unesp.br; 3Centro de Investigación, Experimentación y Servicios del Champiñón (CIES), 16220 Quintanar del Rey, Cuenca, Spain; apardo.cies@dipucuenca.es

**Keywords:** *Agaricus subrufescens*, cereal, food security, sustainable agriculture, mushroom–plant interaction

## Abstract

Cultivating crops in post-harvest areas of sun mushrooms presents an innovative alternative to reduce reliance on mineral fertilizers. Advances in crop cultivation in these areas could make this a sustainable solution for enhancing food security. We evaluated maize cultivation in a sun mushroom post-harvest area, focusing on soil and leaf macronutrient composition, yield factors, and economic benefits. Four management practices were tested: a post-harvest area without mineral fertilization (SMS); a post-harvest area with fertilization at sowing (SMS + S); a post-harvest area with fertilization at sowing and topdressing (SMS + S + TD); and a control area with standard mineral fertilization. The SMS treatment maintained adequate soil pH and electrical conductivity, and in the first crop, increased soil P, Ca, and Mg levels by 5%, 140%, and 23%, respectively, without significantly affecting yield compared to the control. However, nutrient absorption faced challenges due to a nutritional imbalance of Ca/Mg. In the second crop, SMS + S + TD was crucial for higher yields (up to 6500 kg ha^−1^) and showed similarity to the control in a Nearest Neighbor Analysis, particularly in leaf N content. Regarding the economic benefits, SMS reduced mineral fertilization in the first crop, increasing the net benefit by up to 380%, while in the second crop, topdressing became indispensable for the SMS area, with SMS + S + TD generating the greatest net benefit.

## 1. Introduction

Amidst projections of a potential 56% global food demand increase by 2050 [1], the urgency of sustainable farming is intensifying. Coupled with this, events such as pandemics and civil conflicts can increasingly disrupt global food supply chains, particularly affecting smallholders [2,3]. Therefore, it is imperative to utilize Earth’s resources sustainably to ensure food security and mitigate the current impacts of climate change. Among the promising strategies, the diversification of staple crops stands out as a viable option to enhance food security and promote more efficient land use [4].

Conventional crop diversification systems typically include widely cultivated crops such as wheat, rice, corn, barley, forage, and legumes [5]. Among unconventional alternatives, field mushroom cultivation has emerged as a noteworthy option. This approach not only diversifies food production within the same agricultural area, but also fosters a circular economy by repurposing agricultural residues [6,7].

Outdoor mushroom cultivation is particularly notable for its ability to enhance nutraceutical bioactive compounds due to exposure to UV rays [8]. However, studies indicate that mushroom cultivation is seasonally restricted, occupying the land for only a portion of the agricultural year [8]. During this period, it deposits significant amounts of mycelium and organic matter into the soil [8]. These post-harvest residues act as natural soil conditioners, improving soil fertility and promoting plant growth for subsequent crops [9,10]. Therefore, integrating mushroom cultivation into agricultural systems could increase land use efficiency and promote long-term sustainability.

Recent literature demonstrates successful practices of using plants and mushrooms, such as *Volvariella*-coco [11], *Pleurotus*-fava [12], and *Morchella*-peach [13], whether these systems involve cultivating the mushroom prior to or in consortium with vegetable crops. In a specific Brazilian context, small-scale farmers commonly grow sun mushrooms (*Agaricus subrufescens*) in the field [14,15]. However, the following question remains: How can the area be utilized when the climatic conditions are not favorable for mushroom cultivation?

Notably, there is a dearth of scientific data on reusing post-harvest sun mushroom areas. Maize crop offers a strategic solution for reusing sun mushroom post-harvest areas, given its global importance [16] and adaptability [17]. Additionally, maize by-products, such as straw and cobs, can be utilized to produce mushroom substrate [18], promoting a circular economy. Assessing the yield, chemical characteristics, and economic viability of this system can provide valuable insights to increase food security in these regions.

Therefore, we aimed to evaluate the yield of maize cultivation as a function of partial or total replacement of mineral fertilization in a sun mushroom post-harvest area, as well as the effects on the chemical properties of the leaves and soil, and the economic aspects. The purpose of this study was to demonstrate that reusing sun mushroom post-harvest areas for maize cultivation could result in good yields, increase food security, and optimize land use for mushroom farmers.

## 2. Materials and Methods

This study was conducted at the Mushroom Studies Center (CECOG) in the experimental area of the Faculty of Agricultural and Technological Sciences, São Paulo State University (UNESP), Campus Dracena (latitude 21°29′ S and longitude 51°52′ W, average altitude of 420 m), during the period from August 2020 to May 2021. The soil was classified as ARGISSOLO VERMELHO Distrófico according to Santos et al. [19], equivalent to an Ultisol according to Soil Taxonomy [20]. The climate of the region is classified as Cwa [21], with hot and rainy summers from October to March and dry, mild winters with low rainfall from April to September. The local site, monthly average rainfall, and temperature during the experimental period are presented in Figure 1.

Before maize sowing, from April to July 2020, the experimental area was cultivated with sun mushrooms in the open field. Briefly, colonized compost was added to furrows that were approximately 50 cm wide and 30 cm deep, spaced at 60 cm. Details of the mushroom cultivation processes in the field are available in Vieira Júnior et al. [8].

Two maize crops (hybrid K9606VIP3, KWS ^®^ SAAT SE & Co., Einbeck, Germany) were grown consecutively from August 2020 to March 2021, immediately following the mushroom harvest. The maize crop period was strategically planned to cover the entire agricultural year cycle from April 2020 to March 2021. Subsoiling and deep harrowing were performed at a soil depth of 40 cm to ensure uniformity throughout the area, allowing for an even distribution of residual material present in the soil.

Soil samples were collected one day before maize sowing from the control areas (without mushroom cultivation) and treatment areas (with mushroom cultivation) at depths of 0–20 cm and 20–40 cm for soil characterization (Table 1).

### 2.1. Experimental Design

To evaluate the potential of the sun mushroom post-harvest area in maize production, a trial was conducted to compare its use and that of the mineral fertilization techniques recommended in maize production. The treatments carried out were as follows:(i)SMS: sun mushroom post-harvest area without application of mineral fertilizer;(ii)SMS + S: sun mushroom post-harvest area with the application of mineral fertilizer at sowing;(iii)SMS + S + TD: sun mushroom post-harvest area with the application of mineral fertilizer at sowing and topdressing; (iv)Control area following the mineral fertilization recommendations of Duarte et al. [22].

The treatments containing SMS were carried out in the sun mushroom post-harvest area, while the control area was installed close to the plots but at a sufficient distance to not be affected by the post-harvest sun mushroom area (Table 2). Two soil chemical analyses were carried out (control area and post-harvest sun mushroom area).

There were eight replicates for each of the four treatments. The treatments were conducted in blocks with 32 plots, each measuring 1.8 m in width and 5.0 m in length, for a total area of 9 m^2^ per plot. The maize was cultivated with the row spacing fixed at 0.45 m. The population density was set at 66,600 plants per hectare for the initial harvest and 60,000 plants per hectare for the successive crop, in accordance with the manufacturer’s recommendations. These adjustments in the population density were made to account for the specific environmental conditions of each season.

Urea (45% N), triple superphosphate (46% P_2_O_5_), and potassium chloride (60% K_2_O) were used as N, P, and K fertilization sources, respectively. For the experimental plots that had fertilizers applied at sowing (control, SMS + S, and SMS + S + TD), 30 kg ha^−1^ of N, 90 kg ha^−1^ of P_2_O_5_ (in the control area), 50 kg ha^−1^ of P_2_O_5_ (in the experimental area SMS), and 50 kg ha^−1^ of K_2_O were applied. The P_2_O_5_ fertilization in the post-harvest mushroom area was reduced from 90 to 50 kg ha^−1^ due to the higher K content (1.8 mmol_c_ dm^3^) compared to the control area (1.1 mmol_c_ dm^3^), following the fertilization guidelines of Duarte et al. [22] (Table 2).

For the experimental plots receiving topdressing fertilization (control and SMS + S + TD), 60 kg ha^−1^ of N and 20 kg ha^−1^ of K_2_O were added in addition to the sowing fertilizer mentioned above. The topdressing fertilizer was applied twice, at 30 and 45 days after sowing (DAS), and was split applied, with half applied at 30 DAS and the other half at 45 DAS (Table 2).

### 2.2. Maize Cultivation

The first crop was sown in August 2020 and its harvest was carried out in November 2020, while the successive crop sowing was carried out in December 2020 and its harvest was carried out in March 2021. The cultivation cycle lasted 112 days in the first crop and 117 days in the second crop.

Sowing, crop handling, harvesting, and processing were performed according to the methods described by Aguiar et al. [23]. The weeds were removed manually. Micro-sprinklers (4.5 L per hour) were used to irrigate the maize. Irrigation was applied every three days or as needed to address the water deficit during maize cultivation. After physiological maturity, the ears were harvested, and the agronomic aspects were assessed. Grain yield was converted to dry weight with a moisture correction of 13%.

### 2.3. Electrical Conductivity and Soil pH

To evaluate soil electrical conductivity (EC) and pH in response to the post-harvest application of mineral fertilizers in the mushroom area, soil samples were collected from a depth of 0–20 cm between cultivation rows after each maize crop.

To determine EC and pH in the water, a soil-to-water ratio of 1:2.5 (10 cm^3^ of air-dried soil to 25 mL of water) was used. The soil–water mixture was stirred for 30 min, left to settle for another 30 min, and then stirred again for 30 s [24] before measuring EC and pH. An electrical conductivity meter (mCA-150, Tecnopon, Piracicaba, Brazil) and a digital pH meter (mPA-210, Tecnopon) were used for the measurements. The data were evaluated in triplicate.

### 2.4. Chemical Characterization of Soil and Plants

The organic matter (OM) and macronutrient contents of the soil and leaves were evaluated as described by Nogueira et al. [25]. Homogeneous composite samples were collected from three points between cultivation rows in the 0–20 cm layer to measure the OM and macronutrient concentrations in the soil. A total of 32 samples per crop were analyzed for macronutrients such as P, K, Ca, Mg, and S.

To assess the nutritional status of the maize crop, leaves opposite the upper ear were sampled when 50% of the plants were tasseling. Four leaves were collected per plot, and these leaves were combined to form a single composite sample for each plot. The samples were washed with deionized water, dried in an oven with forced air circulation at 65 °C for 72 h, and crushed in a Willey mill. The levels of N, P, K, Ca, Mg, and S were determined from the diagnostic leaf samples, as they are nutritionally representative of the cultivated plants.

### 2.5. Agronomic Parameters

For both crops, eight plants were randomly selected from the two central rows of each experimental plot (with a useful area of approximately 2.5 m^2^ per plot), resulting in a total of 256 plants evaluated per crop. At the tasseling stage (approximately 70 DAS), the following agronomic parameters were measured: plant height (H), stem diameter (D), aboveground dry matter (ADM), the weight of one hundred grains (G100), and grain yield (yield), and the latter was calculated based on the mass of harvested grains per area corrected to 13% moisture.

To assess the levels of chlorophyll a and b and total chlorophyll, a ClorofiLOG portable chlorophyll meter (Falker Automação Agrícola, Porto Alegre, Brazil) was used, which provides the Falker Chlorophyll Index (FCI), proportional to the absorbance of chlorophyll, according to Schlichting et al. [26]. The FCI evaluation was assessed at the tasseling stage on the last fully opened leaf.

### 2.6. Nearest Neighbor and Heat Map Analysis

To analyze the effects of post-harvest treatments and key management on both maize crop cycles, Nearest Neighbor Analysis (NNA) was conducted on the collected agronomic dataset. To visually represent the chemical indices of the soil and leaves, a heat map was selected for use.

### 2.7. Economic Analysis

The economic analysis of maize production follows a partial budget approach [27], aiming to boost the producer’s net crop yield. Net profit (NI) was calculated as the difference between total return (TR), derived from grain sales data [28], and total costs. Fixed costs (FC) encompassed seed, herbicide, pesticide, electricity, labor, and machinery costs. The costs of mineral fertilization were adopted as the variable cost (VC). FC and VC were estimated based on maize family farmers’ production expenses in Brazil in 2020/2021 harvests [29]. All values are presented in USD, with an exchange rate of BRL 4.69.

### 2.8. Statistical Analysis

The results were evaluated separately for each crop cycle (first and second). An analysis of variance (ANOVA) was performed, and the means were compared by Tukey’s test at a 5% probability (R software (v. 4.1.0)). The NNA (Cluster analysis, Euclidean mean) and heat map were assessed by StatGraphics Centurion software (v.16). GraphPad Prism software (version 8.01 (244)) was used to prepare the figures.

## 3. Results

### 3.1. Electrical Conductivity and Soil pH

In the first crop, the values of SMS and SMS + S presented the highest pH values (6.62 and 6.48, respectively), while the treatments that received more fertilizer applications had the lowest pH values (6.25 for control and 6.15 for SMS + S + TD) (Table 3). In the second crop, it was observed that the SMS treatments had a higher pH, with averages between 6.28 and 6.35.

Soil EC exhibited varying effects due to different management practices. In the first harvest, significant differences were observed between SMS + S and SMS + S + TD, with EC increasing by approximately 40.2% following the application of topdressing fertilizer. In the second harvest, the EC values were higher for the treatments that included SMS and SMS + S + TD.

### 3.2. Chemical Characterization of Soil and Plants

All macronutrients varied statistically according to the treatments, except the OM and S content (Table 4). It was observed that the levels of P, Ca, and Mg in the soil showed similar behavior, with SMS presenting the highest value observed in the first crop.

The SMS treatment presented significantly higher values than the control (approximately 5% for P and 140% for Ca). A similar tendency was observed for Mg in the first crop, where SMS increased Mg content by 23% relative to the control. Significant differences in K were only observed in the second harvest, where the SMS + S treatment showed approximately 25% higher K levels compared to the control.

In the second crop, the absence of topdressing fertilizer significantly reduced N content (Table 5). The SMS and SMS + S treatments showed the lowest averages, with 11.55 and 12.25 g of N per kg, respectively. Still, for the second crop, the SMS treatment also presented significantly lower Ca levels (1.20 g of Ca per kg) compared to the control (1.62 g of Ca per kg), while Mg content increased by more than 60% with SMS + S + TD compared to SMS.

### 3.3. Agronomic Parameters

It was observed that FCI increased by more than 22% with the use of SMS + S + TD in the first crop compared to the control (Figure 2). However, in the first crop, there were no significant differences between H, D, and ADM. The successive maize crops in the post-harvest sun mushroom area showed changes in plant characteristics, as significant differences were observed in all the variables mentioned in this section.

Generally, cultivation with SMS in the second crop reduced FCI by about 20% and H by approximately 7% compared to the control. The treatment with SMS + S showed intermediate development, with significant reductions only in FCI, while SMS + S + TD was statistically equivalent to the control for these variables in the second crop (Figure 2).

The importance of optimal fertilizer management is underscored by its direct impact on G100 values and overall yield. In the first harvest, SMS + S maintained G100 values comparable to the control, with an average of 19 g. Additionally, yield can be sustained while reducing mineral fertilizers, as SMS shows no significant differences compared to the control (Figure 3).

In the second crop, the treatment with complete fertilizer application (SMS + S + TD) resulted in higher G100 (26.3 g) and yield (6610.0 kg ha^−1^) values (Figure 3). However, no significant differences were observed in H and D values in the control when fertilizer was applied at sowing (SMS + S) (Figure 2). In particular, treatments without topdressing fertilization (SMS and SMS + S) in the second crop showed lower yields compared to SMS + S + TD, with average yields of 5189 and 5232 kg ha^−1^, respectively (Figure 3).

### 3.4. Cluster and Heat Analysis

The clusters were grouped based on maize treatments and agronomic parameters like FCI, H, D, ADM, G100, and yield. The calorimetric map considered the soil and plant chemical characteristics. Figure 4 shows the results for both crops. The NNA indicated specific groupings; in the first crop, control and SMS + S formed a cluster, while SMS and SMS + S + TD did not (Figure 4a).

The SMS + S treatment had high soil nutrient similarity with the control (Figure 4b). The SMS treatment had elevated soil nutrients with high S absorption (Figure 4c). In the second crop, treatments with topdressing fertilizer (Control and SMS + S + TD) clustered together, highlighting the importance of optimized fertilization (Figure 4d–f).

### 3.5. Economic Analysis

The economic analysis presented in Table 6 sheds light on how fertilization practices can be optimized. It is noted that the control treatment has a variable cost (VC) to the producer of USD 344 compared to the SMS treatment. Despite the lower yield associated with SMS, it results in a remarkable increase in net profit of around 380% in the first crop and 150% in the second crop, underscoring the potential for significant economic gains with relatively low investment post-harvest. As for the SMS + S treatment, it increased yield by 290% in the first crop, but a 4% reduction was observed in the second crop compared to the control. On the other hand, SMS + S + TD initially reduced the net benefit by 50%, while it increased by 170% in the second crop, demonstrating the importance of topdressing fertilization in successive maize crops.

## 4. Discussion

### 4.1. Electrical Conductivity, pH, and Chemical Characterization of Soil and Plants

Maize cultivation in post-harvest areas can be a cost-effective and sustainable alternative to mineral fertilization, due to the presence of organic residues from mushroom cultivation already present in the soil. Moreover, soil pH (6.1 to 6.6) and EC levels (75–159 µS cm^−1^) were found to be within an acceptable range for all treatments and crops, as recommended in the literature [30].

The SMS treatment was better in the first crop because it provided high levels of P, Ca, and Mg in the soil (Table 4 and Figure 4b). However, the main limitation was the uptake of nutrients by the plant. At higher pH levels, the dissolution of phosphate fertilizers decreases, leading to a reduction in plant-available P due to the formation of CaMg apatites [31]

Although SMS is a recognized source of P [31], its low soil mobility hinders P uptake (Table 5, Figure 4c). In the post-harvest sun mushroom area, there was a 40 kg ha^−1^ reduction in P_2_O_5_, which is in line with Duarte et al. [22]. Thus, additional P fertilizers may be unnecessary due to (1) increased demand for P fertilizers; (2) limited investment capacity in Latin American countries such as Brazil; and (3) concentrated sources of mineable P in few countries, emphasizing alternatives phosphorus sources, as organic amendments [32,33].

This reduced availability of nutrients, particularly P, highlights the critical role of Ca and Mg in plant metabolism [34]. Both nutrients are essential for enzyme activation, protein synthesis, photosynthesis, ion transport, and cell division [35]. As divalent cations, Ca and Mg share uptake mechanisms and compete with K. The (Ca + Mg)/K ratio should ideally remain below 30 in crops [36,37]. Our findings, however, reveal ratios of 32 and 39 for SMS and SMS + S treatments, respectively, suggesting the presence of nutritional imbalances (Figure 4c).

In the second crop, topdressing fertilization treatments (control and SMS + S + TD) increased foliar N, enhancing foliar N uptake (Table 4 and Figure 4c) and FCI (Figure 2). SMS + S + TD significantly boosted ADM and yield compared to SMS and SMS + S (Figure 3 and Figure 4a). Furthermore, it was observed that FCI and N followed similar trends in both crops, demonstrating a correlation between N content and chlorophyll.

N fertilizer is crucial for plant growth [38]. SMS initially provides a significant amount of N, but its ammoniacal form (N-NH_4_) eventually surpasses the more easily absorbed nitric form (N-NOx) [39]. Nitrogen’s high mobility and susceptibility to leaching [37,38] require a continuous supply of N. Dunjana et al. [40] emphasize the integration of organic and nitrogen fertilizers to meet the high N demand of maize. This topic is of extreme relevance for smallholder farmers, who face a stagnant global market [41]. Succession planting of maize in a post-harvest sun mushroom area has proven to be affordable; however, N topdressing is necessary to maintain plant nutrition.

### 4.2. Agronomic Parameters

Our research studied maize cultivation in post-harvest sun mushroom areas, with the premise that a model system could be developed that integrates mushrooms and plants for better land use and higher production throughout the year. Despite regional limitations, such as weather conditions and local soil type, the findings of this study may provide an option to increase economic viability, following a premise of better soil conservation and greater food security. This system presents a promising alternative for enhancing soil properties [31], while also serving as a regionally adapted option, given its compatibility with the climatic conditions favorable for sun mushroom cultivation in Brazil [42].

Vieira Júnior et al. [8] underscored in their key findings that sun mushroom production in Brazil could be highly efficient and possess greater bioactive properties only during the summer season, suggesting that cultivation during the remainder of the year should be conducted in controlled environmental settings such as greenhouses and chambers.

The results of our study demonstrated that the SMS treatment (without adding mineral fertilizer to the area after the sun mushroom harvest) resulted in a maize productivity of 5300 kg ha^−1^ in the first crop. Studies by Nunes et al. [43] and Puga et al. [44], reporting yields of 5000 kg ha^−1^ and 5026 kg ha^−1^, respectively, using cattle compost and wood biochar as soil amendments demonstrate that the use of SMS treatment was as effective as the application of other organic amendments.

Specifically, in previous studies, we proved that the spent mushroom substrate increases the agronomic parameters of maize [45], indicating an immediate production option after mushroom harvest, without the necessity of adding minerals. Topdressing in the second crop was crucial to maintain ICF, H, and ADM, since the SMS + S + TD treatment did not show significant differences in relation to the control treatment. The increase in H and ADM with the SMS + S + TD treatment generated greater straw production compared to the SMS and SMS + S treatments. This maize straw can be managed for incorporation into the soil, aiming at increasing carbon [46], or it can be used within a circular economy, with the aim of producing substrate for sun mushrooms [47].

However, it is worth noting that yields and G100 in the SMS and SMS + S treatments did not differ significantly from the control, suggesting that the reduction in biometric characteristics did not impact the final maize yields in the second crop.

### 4.3. Economics Analysis and Implications

Our study aimed to demonstrate, through a simple technique of balancing fixed and variable costs, how the replacement of post-harvest areas can generate economic benefits by reducing the need for mineral fertilizers. This straightforward approach, proposed by Zhai et al. [27], is easy to implement. However, there are limitations that should be considered, such as sensitivity to market fluctuations and input costs.

In a scenario such as that employed in our investigation, the use of post-harvest sun mushroom areas in intercropping systems can increase economic returns, as noted by Song et al. [13]. Technology can improve management for Brazilian smallholders. Brazilian smallholders play an important role in global food security [48].

Our study highlights the potential of post-harvest sun mushroom areas to increase net benefits, especially in the first corn crop. The SMS treatment significantly increased income because no sowing and/or topdressing fertilizers were applied. Abstaining from fertilization reduces costs, enhances energy efficiency, and reduces greenhouse gas emissions. Fathi et al. [49] found that NPK fertilizers are less energy-efficient and raise global warming potential by 15%. Organic sources curtail N_2_O emissions and are more potent than CO_2_ [50].

In the second crop, despite unsatisfactory agronomic results with SMS, it still yielded a USD 152.1/ha increase compared to the control. However, low nutrient absorption posed an issue (Figure 4f). Increasing fertilizer input in the second harvest is economically motivated. Combining sowing and topdressing fertilization notably enhances productivity. Integrating organic and inorganic inputs support enhanced maize productivity in tropical soils [51].

This strategy enables the effective management of land use for year-round agricultural purposes, benefiting Brazilian smallholders by generating income and food security. Combining organic and inorganic fertilization ensures a balanced nutrient supply, as seen in the SMS + S + TD treatment (Figure 4f). Increased ADM production translates into more valuable by-products, crucial for mushroom cultivation substrates [52], promoting a circular economy and minimizing waste.

## 5. Conclusions

In this initial study, we demonstrated that planting maize in a sun mushroom post-harvest area could be a viable alternative to increase grain production during the year for small Brazilian producers. In the first crop, SMS significantly improved soil P, Ca, and Mg levels. In the second crop, the SMS + S + TD combination was essential to maximize yield and nutrient absorption, mainly N in the leaves.

The economic analysis demonstrated that SMS could reduce mineral fertilization costs while increasing net benefits. However, for the second crop, SMS + S + TD emerged as a viable option to enhance soil and plant chemical properties. Long-term trials are needed to understand nutrient cycling between crops from the deposition of this residue in the soil.

## Figures and Tables

**Figure 1 plants-14-01097-f001:**
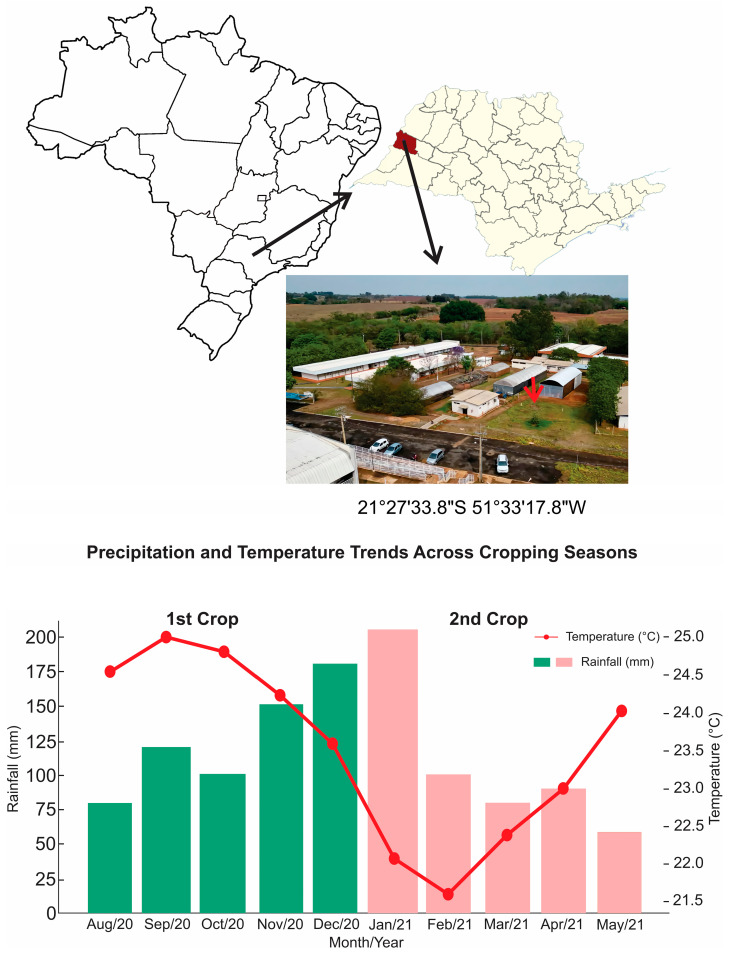
The location of the study area with geographic coordinates (on the **top**) and precipitation (mm) and average temperatures (°C) during the experimental cultivation period of the 1st and 2nd crop seasons in Dracena, SP, Brazil (on the **bottom**). Precipitation and average temperature data were obtained from the meteorological station of FCAT/UNESP (https://www.dracena.unesp.br/#!/estacao-climatologica/, accessed on 1 February 2025).

**Figure 2 plants-14-01097-f002:**
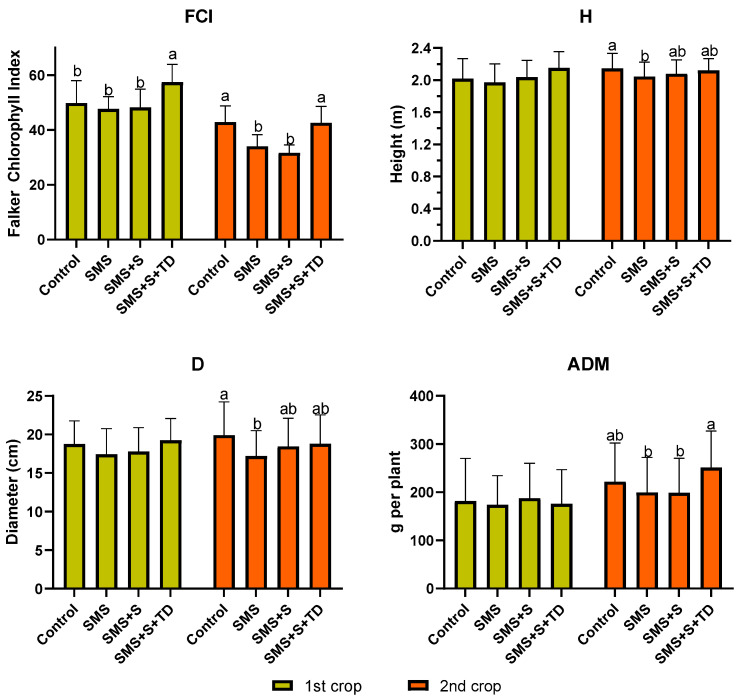
Falker chlorophyll index (FCI), stem diameter (D), plant height (H), and aerial dry matter (ADM) of maize grown in post-harvest sun mushroom area using different management strategies SMS: sun mushroom post-harvest area without application of mineral fertilizer; SMS + S: sun mushroom post-harvest area with application of mineral fertilizer at sowing; SMS + S + TD: sun mushroom post-harvest area with application of mineral fertilizer at sowing and topdressing; Control: area with standard mineral fertilization (sowing and topdressing) following Duarte et al. [22]. Means followed by different lowercase letters between bars of same colors differ according to Tukey’s test (*p* < 0.05).

**Figure 3 plants-14-01097-f003:**
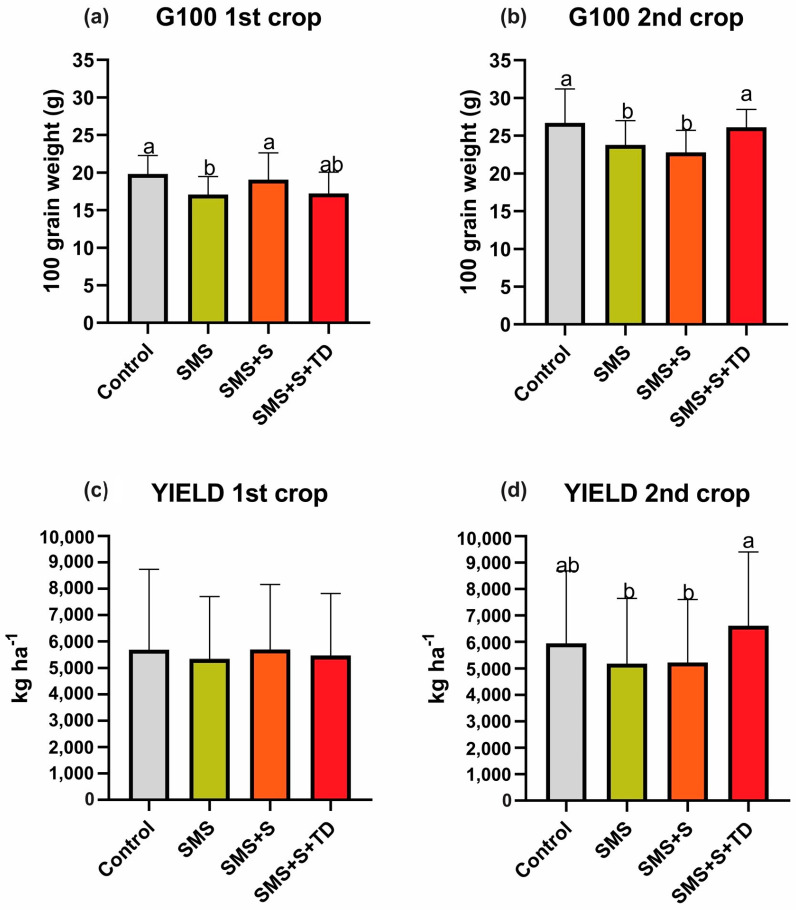
Weight of 100 grains (G100) and yield (YIELD) of maize cultivated in post-harvest sun mushroom area using different management strategies. SMS: sun mushroom post-harvest area without application of mineral fertilizer; SMS + S: sun mushroom post-harvest area with application of mineral fertilizer at sowing; SMS + S + TD: sun mushroom post-harvest area with application of mineral fertilizer at sowing and topdressing; Control: area with standard mineral fertilization (sowing and topdressing) following Duarte et al. [22]. Means followed by different lowercase letters between bars differ according to Tukey’s test (*p* < 0.05).

**Figure 4 plants-14-01097-f004:**
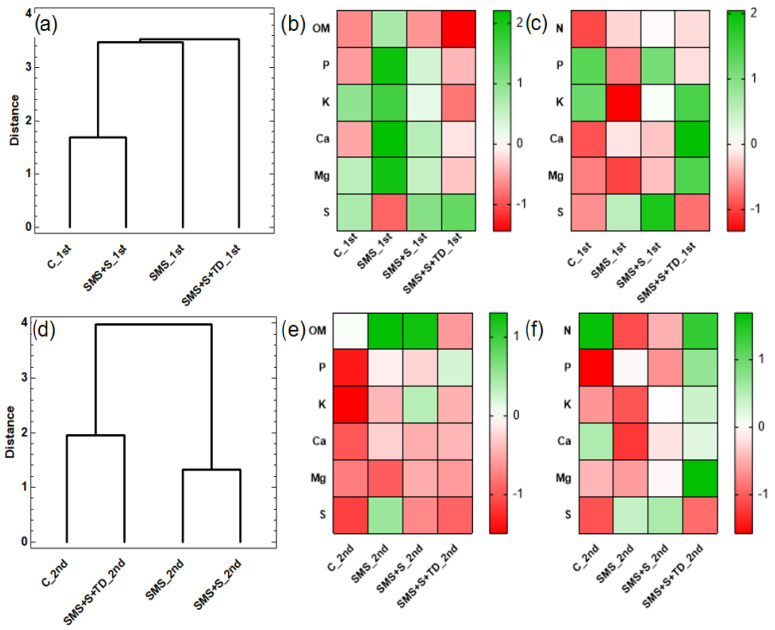
Nearest Neighbor Analysis (Cluster analysis, Euclidean Distance) heat map of soil and leaf organic matter and macronutrients contents for first (**a**–**c**) and second maize crop (**d**–**f**) in sun mushroom post-harvest areas.

**Table 1 plants-14-01097-t001:** The chemical characteristics and granulometry of soil samples collected in the control area and the post-harvest sun mushroom area.

Control Area												
pH	O.M.	P	S	Al^3+^	H + Al	K	Ca	Mg	SB	CEC	BS	AS
CaCl_2_	g dm^3^	mg dm^3^		mmol_c_ dm^3^							%	
5.0	8	4	17	0	16	1.1	16	7	24	40	60	0
Post-harvest sun mushroom area												
pH	O.M.	P	S	Al^3+^	H + Al	K	Ca	Mg	SB	CEC	BS	AS
CaCl_2_	g dm^3^	mg dm^3^		mmolc dm^3^							%	
5.9	9	21	10	0	13	1.8	25	7	34	47	72	0
Soil granulometry												
Clay					Silt				Sand			
g kg^−1^												
130					20				850			

SB: sum of bases; CEC: cation exchange capacity; BS: base saturation; AS: aluminum saturation.

**Table 2 plants-14-01097-t002:** The experimental treatments proposed for the sun mushroom post-harvest area and control area, including the amount of fertilizer and/or the estimated quantity of spent mushroom substrate (SMS) applied to the area.

Area	Treatments	N (kg ha^−1^)	P_2_O_5_ (kg ha^−1^)	K_2_O (kg ha^−1^)	SMS Deposited in the Area (ton ha^−1^)
Post-harvest sun mushroom area	SMS	0	0	0	22.7
	SMS + S	30	50	50	
	SMS + S + TD	90 (30 + 60)	50	70 (50 + 20)	
Control area	Control	90 (30 + 60)	90	70 (50 + 20)	0

SMS: sun mushroom post-harvest area without application of mineral fertilizer; SMS + S: sun mushroom post-harvest area with application of mineral fertilizer at sowing; SMS + S + TD: sun mushroom post-harvest area with application of mineral fertilizer at sowing and topdressing; Control: area with standard mineral fertilization (sowing and topdressing) following Duarte et al. [22].

**Table 3 plants-14-01097-t003:** Mean values for soil pH and electrical conductivity (EC) after maize cultivation (1st and 2nd crops) in sun mushroom post-harvest area.

Treatments	pH (H_2_O)		EC (µS cm^−1^)	
	First Crop	Second Crop	First Crop	Second Crop
Control	6.25 bc	6.14 b	132.37 ab	68.75 b
SMS	6.62 a	6.28 ab	134.50 ab	111.25 a
SMS + S	6.48 ab	6.35 a	95.00 b	75.70 b
SMS + S + TD	6.15 c	6.32 a	158.87 a	101.75 a
Mean	6.38	6.27	130.08	89.36
C.V	3.98	1.97	27.30	20.02
*p* > f	0.001	0.003	0.023	0.000

SMS: sun mushroom post-harvest area without application of mineral fertilizer; SMS + S: sun mushroom post-harvest area with application of mineral fertilizer at sowing; SMS + S + TD: sun mushroom post-harvest area with application of mineral fertilizer at sowing and topdressing; Control: area with standard mineral fertilization (sowing and topdressing) following Duarte et al. [22]. Means followed by different lowercase letters in columns differ according to Tukey’s test (*p* < 0.05).

**Table 4 plants-14-01097-t004:** Mean values for organic matter (OM) and macronutrient (P, K, Ca, Mg, and S) contents after maize cultivation (1st and 2nd crops) in sun mushroom post-harvest area.

Treatments	O.M. (g dm^−3^)		P (mg dm^−3^)		K (mmolc dm^−3^)		Ca (mmolc dm^−3^)		Mg (mmolc dm^−3^)		S (mg dm^−3^)	
	First Crop	Second Crop	First Crop	Second Crop	First Crop	Second Crop	First Crop	Second Crop	First Crop	Second Crop	First Crop	Second Crop
Control	8.04	8.57	4.90 b	4.85 b	1.85	1.22 b	28.71 b	20.95 b	6.82 b	5.27	3.93	2.69
SMS	9.11	9.54	5.07 a	4.93 a	2.02	1.50 ab	71.64 a	32.19 a	8.49 a	5.06	2.86	3.80
SMS + S	8.09	9.48	4.96 ab	4.92 a	1.66	1.69 a	46.46 b	29.09 a	6.73 b	5.59	4.16	2.98
SMS + S + TD	7.45	8.09	4.91 b	4.95 a	1.41	1.48 ab	34.18 b	29.80 a	5.77 b	5.47	4.37	2.83
Mean	8.17	8.92	4.95	4.91	1.73	1.47	45.21	28.00	6.95	5.34	3.83	3.07
C.V	19.33	18.49	1.80	0.63	25.72	22.88	28.21	21.05	17.25	14.52	28.80	28.77
*p* > f	0.023 ^ns^	0.211 ^ns^	0.002	0.000	0.065 ^ns^	0.007	0.001	0.004	0.001	0.562 ^ns^	0.051 ^ns^	0.342 ^ns^

SMS: sun mushroom post-harvest area without application of mineral fertilizer; SMS + S: sun mushroom post-harvest area with application of mineral fertilizer at sowing; SMS + S + TD: sun mushroom post-harvest area with application of mineral fertilizer at sowing and topdressing; Control: area with standard mineral fertilization (sowing and topdressing) following Duarte et al. [22]. Means followed by different lowercase letters in columns differ according to Tukey’s test (*p* < 0.05). ^ns^ = Not Significant.

**Table 5 plants-14-01097-t005:** The mean values for the contents of the macronutrients (N, P, K, Ca, Mg, and S) in the diagnosis of leaves of maize in the sun mushroom post-harvest area.

Treatments	N (g kg^−1^)		P (g kg^−1^)		K (g kg^−1^)		Ca (g kg^−1^)		Mg (g kg^−1^)		S (g kg^−1^)	
	First Crop	Second Crop	First Crop	Second Crop	First Crop	Second Crop	First Crop	Second Crop	First Crop	Second Crop	First Crop	Second Crop
Control	11.72	14.70 a	1.37	1.14	8.65	6.58	1.28	1.62 a	6.17	6.52 b	1.16	1.13
SMS	12.25	11.55 b	1.21	1.26	5.83	6.14	1.46	1.20 b	5.62	6.26 b	1.23	1.22
SMS + S	12.77	12.25 b	1.35	1.21	7.38	7.30	1.42	1.45ab	6.73	7.22 ab	1.21	1.23
SMS + S + TD	12.60	14.35 a	1.25	1.32	8.94	7.71	1.98	1.54 ab	9.74	10.21 a	1.15	1.14
Mean	12.33	13.12	1.30	1.23	7.70	6.93	1.53	1.45	7.09	7.55	1.18	1.18
C.V	15.01	7.15	10.10	11.11	33.87	24.13	39.88	17.63	43.51	33.29	10.00	11.60
*p* > f	0.687 ^ns^	0.000	0.051 ^ns^	0.092 ^ns^	0.091 ^ns^	0.261 ^ns^	0.295 ^ns^	0.015	0.094 ^ns^	0.015	0.461 ^ns^	0.337 ^ns^

SMS: sun mushroom post-harvest area without application of mineral fertilizer; SMS + S: sun mushroom post-harvest area with application of mineral fertilizer at sowing; SMS + S + TD: sun mushroom post-harvest area with application of mineral fertilizer at sowing and topdressing; Control: area with standard mineral fertilization (sowing and topdressing) following Duarte et al. [22]. Means followed by different lowercase letters in columns differ according to Tukey’s test (*p* < 0.05). ^ns^ = Not Significant.

**Table 6 plants-14-01097-t006:** Economics benefits of maize in the post-harvest sun mushroom area using different management strategies.

Crop	Treatments	Total Return			Total Cost		Net Benefit	
		Average Yield (t ha^−1^)	Grain Price (USD t^−1^) ^1^	Total Income (USD ha^−1^)	Fixed Cost (USD ha^−1^) ^2^	Fertilization Cost (USD ha^−1^)	Net Benefit (USD ha^−1^)	Increase (USD ha^−1^) Compared to Control
First Crop	Control	5.64	246.9	1404.9	970.15	344.0	90.7	-
	SMS	5.34	246.9	1318.4	970.15	0	348.2	+257.5
	SMS + S	5.70	246.9	1407.3	970.15	172.0	265.1	+174.4
	SMS + S + TD	5.50	246.9	1357.9	970.15	344.0	43.7	−47.0
Second Crop	Control	5.90	270.5	1595.9	970.15	344.0	281.7	-
	SMS	5.19	270.5	1403.9	970.15	0	433.8	+152.1
	SMS + S	5.23	270.5	1414.7	970.15	172.0	272.5	−9.2
	SMS + S + TD	6.61	270.5	1788.0	970.15	344.0	473.8	+192.1

SMS: sun mushroom post-harvest area without application of mineral fertilizer; SMS + S: sun mushroom post-harvest area with application of mineral fertilizer at sowing; SMS + S + TD: sun mushroom post-harvest area with application of mineral fertilizer at sowing and topdressing; Control: area with standard mineral fertilization (sowing and topdressing) following Duarte et al. [22] ^1^ Values obtained from average grain price in November 2020 and March 2021 in CEPEA/ESALQ; ^2^ estimated average cost of grain production in 2020/2021 harvest for family farmers according to CONAB. Exchange rate was calculated to be BRL 4.69.

## Data Availability

Data are contained within the article.

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
