# Peer review of "Maize Cultivation in Sun Mushroom Post-Harvest Areas: Yield, Soil Chemical Properties, and Economic Viability"

_plants, 2025, doi:10.3390/plants14071097_

Round 1

Reviewer 1 Report

Comments and Suggestions for Authors

The manuscript is in general written in a clear way, easy to read but some sentences needs improvement, and with appropriate reference to data in tables and figures. The work requires corrections. The purpose of the work should be better articulated.

  1. ABSTRACT

- The abstract  requires thorough editing. Generaly abstract should contain the most important information about the research being conducted, especially the purpose of the research.

Keywords should be improved.

  1. INTRODUCTION

Authors should specify the purpose of the work in this part.  Therefore research hypothesis should be formulated.

MATERIAL AND METHODS

- this part needs major improvement because it is not clear.

- there is no detailed description of what the research factor was and what specific levels of the factor were used, determining them will make it easier for the authors to formulate a summary;

RESUTS and DISCUSSION

the results and disusssion should be analysed for differences in the test series and in the analyses performed so as to present actual values on the graphs.

The discussion needs to be re-examined by the authors because, especially in the first part related with Agronimic parameters, many sentences are not understandable.

All my suggestions are included in the text.

Author Response

Reviewer 1

The manuscript is in general written in a clear way, easy to read but some sentences needs improvement, and with appropriate reference to data in tables and figures. The work requires corrections. The purpose of the work should be better articulated.

Answer: The authors appreciate all comments and suggestions provided for the article. All suggested changes have been incorporated and are highlighted in yellow in the manuscript. Details of each suggestion, along with the response to each request, are provided below.

ABSTRACT

- The abstract  requires thorough editing. Generaly abstract should contain the most important information about the research being conducted, especially the purpose of the research.

Answer: The authors appreciate the comments and emphasize that the requested suggestions have been incorporated into the new version of the manuscript (lines 16-32)

Keywords should be improved.

Answer: The authors appreciate the comments and emphasize that the requested suggestions have been incorporated into the new version of the manuscript (lines 33-34)

INTRODUCTION

Authors should specify the purpose of the work in this part.  Therefore research hypothesis should be formulated.

Answer: The authors appreciate the comments and emphasize that the requested suggestions have been incorporated into the new version of the manuscript (lines 69-72)

MATERIAL AND METHODS

- this part needs major improvement because it is not clear.

- there is no detailed description of what the research factor was and what specific levels of the factor were used, determining them will make it easier for the authors to formulate a summary;

Answer: The authors appreciate the comments and emphasize that the requested suggestions have been incorporated into the new version of the manuscript (section material and methods, 2.1 Experimental Design; 2.2 Maize cultivation; 2.3 Electrical Conductivity and Soil pH, 2.4 Chemical Characterization of Soil and Plant, 2.5. Agronomic Parameters)

RESUTS and DISCUSSION

the results and disusssion should be analysed for differences in the test series and in the analyses performed so as to present actual values on the graphs.

Answer: The authors appreciate the comments and emphasize that the requested suggestions have been incorporated into the new version of the manuscript (lines 301-307)

The discussion needs to be re-examined by the authors because, especially in the first part related with Agronimic parameters, many sentences are not understandable.

Answer: The authors appreciate the comments and emphasize that the requested suggestions have been incorporated into the new version of the manuscript (Section 4.2 Agronomic Parameters)

All my suggestions are included in the text.

Reviewer 2 Report

Comments and Suggestions for Authors

The article is very interesting and addresses a very pertinent and timely topic, given the current global panorama.

Although english is not my native language, it seams to me that could be improved for a more clear reading and understanding of the meaning of the sentences.

In the Introduction, the second paragraph is very nuclear. From line 40 to 43, I cannot even understand what the authors want to say. Not only is it not clear, but I do not agree with what I think is meant, at least as it is.

I think the Material and methods chapter is a bit confusing, it should combine the evaluations and analyses carried out on the plants and separate them from the soils, in addition, the text could be clearer, and there are several issues to be highlighted, namely:

  • - Line 81 to 82 - The caption of figure 1 is unclear and does not provide the necessary information (especially with regard to meteorological data);
  • - Line 95 – the caption is poorly written, it should be “Chemical characteristics and granulometry of soil samples collected in…”;
  • Line 97 to 98 – There is no need to specify here the method for P extraction, it has to be in the texto, also, the uppercase letters before the acronyms are not necessary, just write the acronym and identify it;
  • Line 100 to 113 – These 2 paragraphs should be rewritten so as not to repeat any of the information. I suggest changing the order of the paragraphs and reviewing what is written;
  • Line 111 - must specify what the control treatment consists of;
  • Line 122 to 128 – This text should be integrated into the paragraph describing how the treatments were carried out and not at the bottom of the table. What should be placed here is the identification of the acronyms used in the table;
  • Line 140/141 – I suggest you write “The weeds were removed manually”, instead “Manual weeding was used for weed control.”;
  • Line 143 to 144 – explain what do you mean, please;
  • Line 158 to 164 - says that it makes assessments (why just EC and pH?), but does not say when, or why, in addition, just below it talks about the characterization of the soil (see initial observation made to this chapter, please);
  • Line 165 on – When were the samples taken and how;
  • Line 173 to 175 - specify the characteristics of the wash water and how it was made, and you don’t need to say that the samples were stored.

Regarding the Results, it would be important for them to appear in the same order as they are mentioned in the previous chapter, causing less confusion. Therefore, it should start by referring to the results of the plant measurements, then moving on to the physiological evaluations. The text is a little confusing, as it refers to one characteristic, then presents results from another, then returns to the previous one, making the text not easy to read.

Figure captions should be reviewed, because they contain errors (see the acronyms ADM and G100).

In addition to the previous observations, there are some aspects to review, namely:

  • - Line 201 to 203 - I don't think there is any need for this sentence, besides it doesn't seem to be well structured;
  • Line 221 – It is not correct to use the expression “statistically equal”, instead it should say that it “was not significantly different”, and therefore, it should not say what follows, in line 222;
  • Line 224 – the average yield should be presented as one value alone and not “between” (as they do not differ significantly);
  • Line 243 – I would suggest to change the text because the measurements, as I understand, were made after maize harvest, or were not, the text should be clear, it shouldn't raise so many doubts;
  • Line 249 - It should be stated here in the text that the results refer to the soils, making it clear when the samples were taken, and I do not see values for “microminerals” in the table (I think you meant “macronutrients”);
  • Line262 – SEE THE COMMENT MADE FOR LINE 249.
Comments on the Quality of English Language

The english must be improved

Author Response

Reviewer 2

The article is very interesting and addresses a very pertinent and timely topic, given the current global panorama.

Although english is not my native language, it seems to me that could be improved for a clearer reading and understanding of the meaning of the sentences.

Answer: The authors appreciate all comments and suggestions provided for the article. All suggested changes have been incorporated and are highlighted in yellow in the manuscript. Details of each suggestion, along with the response to each request, are provided below.

In the Introduction, the second paragraph is very nuclear. From line 40 to 43, I cannot even understand what the authors want to say. Not only is it not clear, but I do not agree with what I think is meant, at least as it is.

Answer: The authors appreciate the comments and emphasize that the requested suggestions have been incorporated into the new version of the manuscript (Lines 41-56)

I think the Material and methods chapter is a bit confusing, it should combine the evaluations and analyses carried out on the plants and separate them from the soils, in addition, the text could be clearer, and there are several issues to be highlighted, namely:

  • Line 81 to 82 - The caption of figure 1 is unclear and does not provide the necessary information (especially with regard to meteorological data);

Answer: The authors appreciate the comments and emphasize that the requested suggestions have been incorporated into the new version of the manuscript (Figure 1, Line 79 and 86)

  • Line 95 – the caption is poorly written, it should be “Chemical characteristics and granulometry of soil samples collected in…”;

Answer: The authors appreciate the comments and emphasize that the requested suggestions have been incorporated into the new version of the manuscript (Line 115)

  • Line 97 to 98 – There is no need to specify here the method for P extraction, it has to be in the texto, also, the uppercase letters before the acronyms are not necessary, just write the acronym and identify it;

Answer: The authors appreciate the comments and emphasize that the requested suggestions have been incorporated into the new version of the manuscript (Line 117)

  • Line 100 to 113 – These 2 paragraphs should be rewritten so as not to repeat any of the information. I suggest changing the order of the paragraphs and reviewing what is written; Line 111 - must specify what the control treatment consists of;

Answer: The authors appreciate the comments and emphasize that the requested suggestions have been incorporated into the new version of the manuscript (Lines 119-150)

  •  
  • Line 122 to 128 – This text should be integrated into the paragraph describing how the treatments were carried out and not at the bottom of the table. What should be placed here is the identification of the acronyms used in the table;

Answer: The authors appreciate the comments and emphasize that the requested suggestions have been incorporated into the new version of the manuscript (Table 2, Lines 137-143)

  • Line 140/141 – I suggest you write “The weeds were removed manually”, instead “Manual weeding was used for weed control.”;

Answer: The authors appreciate the comments and emphasize that the requested suggestions have been incorporated into the new version of the manuscript (Line 170)

  • Line 143 to 144 – explain what do you mean, please;

Answer: The authors appreciate the comments and emphasize that the requested suggestions have been incorporated into the new version of the manuscript (Line 170-172)

  • Line 158 to 164 - says that it makes assessments (why just EC and pH?), but does not say when, or why, in addition, just below it talks about the characterization of the soil (see initial observation made to this chapter, please) - Line 165 on – When were the samples taken and how;

Answer: The authors appreciate the comments and emphasize that the requested suggestions have been incorporated into the new version of the manuscript (Line 176-178)

  •  
  • Line 173 to 175 - specify the characteristics of the wash water and how it was made, and you don’t need to say that the samples were stored.
  • Answer: The authors appreciate the comments and emphasize that the requested suggestions have been incorporated into the new version of the manuscript (Line 192-195)

Regarding the Results, it would be important for them to appear in the same order as they are mentioned in the previous chapter, causing less confusion. Therefore, it should start by referring to the results of the plant measurements, then moving on to the physiological evaluations. The text is a little confusing, as it refers to one characteristic, then presents results from another, then returns to the previous one, making the text not easy to read.

Answer: The authors appreciate the comments and emphasize that the requested suggestions have been incorporated into the new version of the manuscript. All sections have been changed, from materials and methods, results and discussion, as well as the order and numbering of references, for better understanding.

Figure captions should be reviewed, because they contain errors (see the acronyms ADM and G100).

In addition to the previous observations, there are some aspects to review, namely:

  • Line 201 to 203 - I don't think there is any need for this sentence, besides it doesn't seem to be well structured;

Answer: The authors appreciate the comments and emphasize that the requested suggestions have been incorporated into the new version of the manuscript (Line 277)

  • Line 221 – It is not correct to use the expression “statistically equal”, instead it should say that it “was not significantly different”, and therefore, it should not say what follows, in line 222;

  • Answer: The authors appreciate the comments and emphasize that the requested suggestions have been incorporated into the new version of the manuscript.
  • Line 224 – the average yield should be presented as one value alone and not “between” (as they do not differ significantly); Line 243 – I would suggest to change the text because the measurements, as I understand, were made after maize harvest, or were not, the text should be clear, it shouldn't raise so many doubts;

Answer: The authors appreciate the comments and emphasize that the requested suggestions have been incorporated into the new version of the manuscript. (Lines 301-306)

  •  
  • Line 249 - It should be stated here in the text that the results refer to the soils, making it clear when the samples were taken, and I do not see values for “microminerals” in the table (I think you meant “macronutrients”); Line262 – SEE THE COMMENT MADE FOR LINE 249.

Answer: The authors appreciate the comments and emphasize that the requested suggestions have been incorporated into the new version of the manuscript. (Lines 249 and Table 4)

Round 2

Reviewer 1 Report

Comments and Suggestions for Authors

The authors substantially improved the text of manuscript. I have no more comments. The manuscript can be accepted by the Publisher.

Author Response

The authors sincerely appreciate your valuable contribution and the improvements you have brought to our work.

Reviewer 2 Report

Comments and Suggestions for Authors

I appreciate the attention paid to the repairs made to the previous version, however, there are still some details that need to be corrected, namely:

 Line 119 to 122 - This text is not necessary, it is part of the introduction and justification of the present study, here it should only be described what was done;

Line 133 - treatments are not “installed”, they are “carried out”

Line 193 - in the previous version of the article it was said that the samples were washed, and very well, however, it should have said that they were washed with deionized water (if that was the case), removing this observation from the text was not the idea, I think it should be included, because it is an important step, to avoid possible errors in laboratory determination, that’s what we do;

Line 241 – In the table you present “Mean values for soil pH and electrical condictivity (EC) for each treatment, in the end of each maize cycle (1st and 2nd crop)” and that’s what should be written; This observation also applies to the remaining tables where the results are presented, which are “Mean values of…”;

Line 289 – the caption of figure 2 refers to G100 and the figure shows ADM;

Line 308 - the caption of figure 3 refers to ADM and the figure shows G100.

Comments on the Quality of English Language I'm not an English expert, however, the work seems to be fine to me

Author Response

We sincerely appreciate your valuable comments and suggestions, which have helped us improve the quality and clarity of our manuscript. Below, we address each of your observations:

L119 - We acknowledge that the previous text contained elements of introduction and justification, following the recommendations of Reviewer 1. This section has been revised while ensuring that it remains a methodological description rather than an introductory statement.

L133- We have corrected the wording, replacing "installed" with "carried out" 

L193- This detail has been reintroduced into the revised version of the manuscript.

L241- The wording in the table captions has been revised.  This correction has also been applied to all other tables presenting results.

L289 and L308- We have corrected the captions of Figures 2 and 3.

We appreciate your careful review and believe that these revisions improve the manuscript's clarity and accuracy. Please let us know if further adjustments are needed.